# Examining isokinetic knee peak torque and time to peak torque as predictors of vertical jump height in division I men's basketball players

Caroline Westwood[1], Arakua Welbeck[2], Carolyn Killelea[3], Peter Howard[4],
Mallory Faherty[5], Daniel Le[6], Ryan Zerega[7], Charles R. Reiter[8], Timothy C. Sell[7]*

1 Lewis Katz School of Medicine, Temple University, Philadelphia, Pennsylvania, United States of America, 2 Washington University of St. Louis School of Medicine, St. Louis, Missouri, United States of America, 3 Duke Department of Orthopedics, Duke University, Durham, North Carolina, United States of America, 4 George Washington University School of Medicine and Health Sciences, Washington, District of Columbia, United States of America, 5 OhioHealth Research Institute, OhioHealth, Columbus, Ohio, United States of America, 6 Michael W. Krzyzewski Human Performance Lab, Department of Orthopaedic Surgery, Duke University, Durham, North Carolina, United States of America, 7 Atrium Health Musculoskeletal Institute, Atrium Health, Charlotte, North Carolina, United States of America, 8 VCU School of Medicine, Richmond, Virginia, United States of America

* timothy.sell@atriumhealth.org

## Abstract

The vertical jump (VJ) is one of the most important movements for basketball athletes and therefore determining modifiable predictors of the VJ would aid substantially in crafting more effective training regimens. The purpose of this study was to determine if isokinetic quadriceps strength and torque predict VJ height and which characteristics and testing speed is the strongest predictor of VJ height. Fifteen subjects (age: $18.5 \pm 1.0$ years, height: $195.9 \pm 6.9$ cm, weight: $96.2 \pm 13.7$ kg) from a single Division I men's basketball team were recruited for this study. All participants performed a standing vertical jump with arm-swing to assess their maximum VJ height. Participants also completed an isokinetic knee extension strength protocol that included testing at multiple speeds. Pearson and Spearman tests found no significant correlation between jump height and peak torque at any of the speeds. Regression analysis showed a statistically significant relationship between time to peak torque at 300°/s and VJ height ($R^2 = 0.23$, $p = 0.04$). These findings suggest that in a population of elite basketball players, the knee's ability to rapidly generate torque likely plays a greater role in VJ performance than its ability to generate high magnitude of torque. This presents a potential benefit of explosive training regimens such as plyometrics for maximization of jump performance.

## 1. Introduction

In basketball, few abilities affect as many facets of the game as the vertical jump (VJ). With this in mind, strength and conditioning coaches at all levels dedicate considerable time and effort towards its improvement, utilizing various training modalities in the process [1]. There

**Data availability statement:** All other relevant data are within the paper.

**Funding:** The author(s) received no specific funding for this work.

**Competing interests:** The authors have declared that no competing interests exist.

has been extensive research on the VJ's biomechanics; however, there are still gaps in the knowledge of what contributes to a good VJ, especially in elite basketball players [2]. While it is well-known that strength of the knee joint, especially in extension, plays a role in vertical jump performance [3–5], details such as which strength parameters best correlate with VJ height remain open to debate.

The relationship between VJ performance and isokinetic knee strength has been examined with several studies demonstrating a moderate to strong correlation between extension peak torque (PT) and VJ height [6,7]. However, two studies have reached the opposite conclusion as well, failing to find a significant correlation between isokinetic PT and VJ height [8,9]. In particular, Alemdaroğlu found no relationship between knee extension peak torque and VJ height in high level Turkish basketball players [10]. Generating a high magnitude of torque about the knee joint appears to be important, but it is not the only important characteristic of isokinetic knee strength. Multiple studies have highlighted the importance of the knee extensors' capacity for rapid torque generation [11–13], demonstrating that a decreased time to peak torque has been correlated with improved VJ performance [11]. These two variables, extension PT and time to PT, represent contractile strength and speed respectively. A comparison of the effect of PT and time to PT on VJ height is therefore asking whether maximal force or the ability to rapidly generate force is more important in maximizing VJ height. Additionally, there is evidence that the speed of isokinetic testing plays a role in its correlation to VJ [14,15].

Ultimately, the gaps in the literature highlight the need for further research examining the relationship between knee strength and VJ performance, particularly in a population of Division I basketball players. It was hypothesized that of the tested parameters, time to peak torque at a speed of 300 °/s will correlate most strongly with VJ height. The results of this study will contribute to better understanding of the relationship between muscular contractile properties and VJ performance and impact the training strategies for athletes in sports that require jumping.

## 2. Methods

### 2.1 Experimental approach to the problem

The objective of this study was to determine the relationship between knee strength, time to peak torque, and VJ height. This was assessed using an isokinetic dynamometer to measure isokinetic knee strength at 60, 180, and 300 degrees and a vertical jump task to measure peak jump height.

### 2.2 Subjects

National Collegiate Athletic Association Division I male basketball players from a single university were recruited for participation in this study. A total of 15 NCAA Division I Men's Basketball athletes participated in the study (Table 1). Testing was conducted during the summer prior to the start of each basketball season from July 1 2017 to July 12 2019. All subjects were cleared by medical personnel (certified athletic trainer and/or team physician)

**Table 1. Population demographics.**

|             | N  | Mean   | SD    |
|-------------|----|--------|-------|
| Age (yrs)   | 15 | 18.53  | 1.02  |
| Height (cm) | 15 | 195.95 | 6.90  |
| Weight (kg) | 15 | 96.24  | 13.67 |

to participate in full physical activity. Subjects were excluded if they did not meet medical clearance. All subjects provided written informed consent as approved by Duke University's Institutional Review Board. If subjects were minors at the time of the study, verbal informed consent was obtained from parents.

## 2.3  Procedures

This study is a descriptive cross-sectional study exploring correlation between two isokinetic knee strength parameters (extension peak torque normalized to body weight and knee extension time to peak torque) and vertical jump height.

An isokinetic dynamometer (Biodex Medical Systems, Inc., Shirley, NY, USA) was used to assess isokinetic knee strength bilaterally. The dynamometer was set up and calibrated according to manufacturer specifications. Knee flexion and extension strength testing was performed with the subject seated and strapped in at the torso, thigh, and ankle. The isokinetic dynamometer point of rotation was aligned with the subject's knee joint. Range of motion limits were set with maximum flexion measured at slightly past 90 degrees and maximum extension measured at about zero degrees. Testing protocol was a concentric-concentric protocol. Testing was performed at three speeds: 60°/s, 180°/s, and 300°/s. Subjects were instructed to kick out and back as hard and as fast as possible. Before each trial, subjects were given a practice of 6 repetitions, 3 at half effort and 3 at full effort. The measured trial consisted of 5 repetitions at 60°/s, 10 repetitions at 180°/s, and 15 repetitions at 300°/s, all at maximal effort. The subjects were given a 30 second rest period between each speed. Trials were excluded if the subject performed the task incorrectly or if the task ended prematurely due to pain. Extension peak torque/BW and time to peak torque were computed as averages of each recorded repetition across both legs.

Vertical jump height testing was performed using a vertical jump testing apparatus (Vertec, Sports Imports, Hilliard OH). Subjects performed a standing countermovement jump with arm swing. Subjects were allowed more attempts as long as they continued hitting higher marks. The subject would receive no more attempts if he failed to reach a higher mark in two subsequent attempts. Vertical jump height was calculated as the difference between the highest mark the subject reached and standing reach. Standing reach was measured by having each subject stand and raise their arm against a wall.

## 2.4  Statistical analysis

All statistical analyses were done in IBM SPSS 24 (IBM Corp.; Armonk, NY). First, descriptive data (means, medians, minima, maxima, standard deviations) were calculated. A Shapiro-Wilk test was done to assess for normality. In order to examine the relationships between the strength parameters and VJ, Pearson and Spearman correlations were calculated for normal and non-normal distributions, respectively. A stepwise multiple linear regression model was fit using the strength variables to predict VJ height. Significance was set *a priori* at $p < 0.05$.

## 3.  Results

Table 2 includes the descriptive data for averages of the subjects' performance in isokinetic dynamometry and VJ. As the angular velocity increased, there was a decrease in both peak torque and time to peak torque. VJ height ranged from 57.15 to 87.63 centimeters with a mean of 76.37 and standard deviation of 18.44.

The correlations between the strength variables and VJ performance are presented in Table 3. There were no significant correlations observed.

**Table 2. Average Isokinetic dynamometry and VJ performance.**

| Variable | Mean | SD | Min. | Max. | Median |
|---|---|---|---|---|---|
| 60°/s E Peak torque/BW (Nm/kg) | 286.25 | 60.31 | 197.75 | 398.87 | 277.78 |
| 60°/s E time to PK TQ (ms) | 447.33 | 149.94 | 275.00 | 790.00 | 385.00 |
| 180°/s E Peak torque/BW (Nm/kg) | 209.26 | 24.14 | 164.05 | 248.33 | 211.04 |
| 180°/s E time to PK TQ (ms) | 213.33 | 41.78 | 135.00 | 285.00 | 220.00 |
| 300°/s E Peak torque/BW (Nm/kg) | 164.04 | 13.64 | 133.15 | 183.81 | 164.65 |
| 300°/s E time to PK TQ (ms) | 165.33 | 49.95 | 115.00 | 290.00 | 150.00 |
| VJ (cm) | 76.37 | 6.83 | 57.15 | 87.63 | 77.47 |

Time (ms).

Abbreviations: E, extension; BW, body weight (kg); PK TQ, peak torque (Nm).

**Table 3. Correlations between Isokinetic dynamometry variables and VJ height.**

| Strength Variable | Correlation Coefficient (R) | p-value |
|---|---|---|
| 60°/s E Peak torque/BW (Nm/kg) | −0.102 | 0.717[1] |
| 60°/s E time to PK TQ (ms) | 0.050 | 0.859[2] |
| 180°/s E Peak torque/BW (Nm/kg) | 0.077 | 0.785[1] |
| 180°/s E time to PK TQ (ms) | 0.048 | 0.864[1] |
| 300°/s E Peak torque/BW (Nm/kg) | 0.208 | 0.456[1] |
| 300°/s E time to PK TQ (ms) | −0.440 | 0.101[2] |

[1]Pearsons.

[2]Spearmans.

**Table 4. Regression between Isokinetic Dynamometry variables and VJ height.**

| Source | SS | df | MS | Observations | 15 |
|---|---|---|---|---|---|
| **Model** | 31.04 | 1 | 31.04 | **F(1,13)** | 5.18 |
| **Residual** | 77.89 | 13 | 5.99 | **Prob > F** | 0.0404 |
| **Total** | 108.93 | 14 | 7.78 | **R²** | 0.285 |
| | | | | **Adjusted R²** | 0.2300 |
| **Predictor Variables** | **Coefficient** | | **t** | **p-value** | |
| Extension time to peak torque 300°/s | -0.0288 | | -2.28 | 0.040 | |
| Constant | 34.8285 | | 15.94 | 0.000 | |

Abbreviations: SS, sum of the squares; df, degrees of freedom; MS, mean squares.

The regression analysis results are presented in Table 4 below. The overall model was significant (p = 0.0404) but only included a single predictor variable (knee extension time to peak torque at 300°/s). The final model explained 23% of the variance in vertical jump performance.

## 4. Discussion

The purpose of this study was to determine if isokinetic knee strength, including time to peak torque, predicts vertical jump performance. It was hypothesized that time to peak torque at the highest speed of 300 °/s would have the strongest correlation with VJ performance.

Overall, the results of this study did not support this hypothesis, as none of the predictor variables showed a significant correlation with VJ height. However, the regression analysis did demonstrate that knee extension time to average peak torque at 300 °/s was a significant predictor of VJ performance and explained 23.0% of the variance in VJ performance. These results should provide clinicians and performance coaches important information for training. Specifically, the development of explosive strength, and thus decreasing the time to peak torque, may improve jump performance.

The only identified predictor variable of VJ height was extension time to average peak torque at 300 °/s (5.23 rad/sec). As knee joint angular velocity can approach 500 °/s during a VJ [16], tasks that approach these speeds may more closely predict performance. The regression analysis also corresponds with Tisokanos et al, which found low relationships between jumping height and hip, knee, and ankle torques with regression beta coefficients from 0.035-0.056 [7]. Previous studies by Iossifidou et al which compared isokinetic strength testing at 30, 90, 180, and 300 degrees with vertical jump performance showed that isokinetic knee extension testing becomes a better predictor of VJ as the angular velocity during testing approaches the angular velocity of the knee joint during a VJ [14]. A study by Young et al in 29 male recreational and club-level athletes drew similar conclusions that speed-strength capacity of leg extensors correlated significantly with jumping performance while maximum strength was not significant [17]. These findings may suggest that the ability to rapidly generate torque is a statistically more important determinant of VJ height than maximal torque production.

This experiment failed to find a significant correlation between peak torque and VJ height, and this may be explained by the role of the hip joint during the VJ. The findings of this experiment were similar to that of Alemdaroglu, who found no correlation between the knee's isokinetic peak power and vertical jump height in elite basketball players [10]. This similarity in results and population suggest that basketball athletes may shift the load of vertical jump generation away from the knee joint. Biarticular muscles, such as semimembranosus, rectus femoris, semitendinosus, and biceps femoris, also play a role in vertical force generation and therefore, their effects on hip joint need to be taken into account [14]. This can be alternatively explained by a different study that found there is a significant inverse relationship between power generated by the knee and power generated by the hip during the VJ, leading to the conclusion that individuals choose to emphasize either the hip or the knee in the VJ [18]. In a separate paper, Vanezis and Lees found that as VJ height increased, only effort exerted by the hip increased significantly, while effort of the knee and ankle remained relatively unchanged [19]. Overall, this may suggest that the tasks required of elite basketball players to increase jump height past a certain threshold involve an emphasis of the hip rather than the knee for additional force generation.

This study has several limitations. A main source of variation in data is the use of an arm swing in jumps. Participants were not given specific instructions on upper body movements during the jump task. The current study allowed the subjects to use their arms/hands during the jumps which was necessary based on the use of the Vertec. This could have affected the ability to find correlations between jump height and knee extensor strength. Previous studies by Hughes et al found significant differences in jump height, peak vertical ground reaction force, and peak concentric power in countermovement jumps with versus without arm swings [20]. Additionally, a larger sample size would have been beneficial in augmenting statistical power of the analyses and increasing the study's validity. This study also only analyzed data from elite male college basketball players, who would not necessarily be generalizable to other athletic populations or other teams not following the same training regimens.

## 5. Practical applications

This data suggests that training regimens should focus on rapid generation of torque in addition to traditional strengthening. Focusing on explosive movements, such as plyometrics [21] or Olympic weightlifting [22], might be worth adding if the goal is improvement of VJ performance. A study of 41 adolescent male soccer players assessed free-weight resistance training, plyometric training, and no additional conditioning throughout an 8-week period. Both free-weight resistance training and plyometric training increased peak torque and rate of torque development. Specifically, plyometric training resulted in greater dynamic power improvement in fast contractions (300 deg/sec) which resulted in increased jump performance [23]. The results of these previous studies and this experiment suggest that relatively less focus should be given to regimens that focus on heavy weightlifting, though it does not make a case for its complete exclusion. Further research should focus on the contribution of other joints and muscles to VJ height, particularly the hip and biarticular muscle groups that cross both the hip and knee joints. Furthermore, comparison of training programs that include plyometric alone versus plyometric and strength training could help determine the true significance of plyometric training for jump performance.

Overall, this study found that the only significant predictor of VJ performance was knee extension time to average peak torque at 300 degrees/second, which suggests explosiveness and rapid generation of torque could be utilized to improve jump performance. These findings emphasize the importance of training regimens focused on the fast recruitment of muscle groups to rapidly generate high levels of torque and explosive training such as plyometrics.

## Acknowledgements

The authors thank all participants for their contributions, involvement, and willingness to partake.

## Author contributions

**Data curation:** Caroline Westwood, Arakua Welbeck.

**Formal analysis:** Carolyn Killelea, Mallory Faherty, Ryan Zerega, Timothy C. Sell.

**Investigation:** Caroline Westwood, Peter Howard, Mallory Faherty, Daniel Le, Ryan Zerega, Charles R. Reiter, Timothy C. Sell.

**Supervision:** Carolyn Killelea, Mallory Faherty, Timothy C. Sell.

**Writing – original draft:** Caroline Westwood.

**Writing – review & editing:** Caroline Westwood, Arakua Welbeck, Carolyn Killelea, Mallory Faherty, Timothy C. Sell.

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
