## [Decision Letter · Decision Letter 0]

16 Nov 2023

PONE-D-23-32453Examining Isokinetic Knee Peak Torque and Time to Peak Torque as Predictors of Vertical Jump Height in Division I Men’s Basketball PlayersPLOS ONE

Dear Dr. Westwood,

Thank you for submitting your manuscript to PLOS ONE. After careful consideration, we feel that it has merit but does not fully meet PLOS ONE’s publication criteria as it currently stands. Therefore, we invite you to submit a revised version of the manuscript that addresses the points raised during the review process.

**ACADEMIC EDITOR: **Dear Authors,your manuscript has been revised by one expert in the field that retrieved several major concerns you should consider during the revision process. Please submit your revised manuscript by Dec 31 2023 11:59PM. If you will need more time than this to complete your revisions, please reply to this message or contact the journal office at plosone@plos.org . Please include the following items when submitting your revised manuscript:

We look forward to receiving your revised manuscript.

Kind regards,

Emiliano Cè

Academic Editor

PLOS ONE

2. PLOS requires an ORCID iD for the corresponding author in Editorial Manager on papers submitted after December 6th, 2016. Please ensure that you have an ORCID iD and that it is validated in Editorial Manager. To do this, go to ‘Update my Information’ (in the upper left-hand corner of the main menu), and click on the Fetch/Validate link next to the ORCID field. This will take you to the ORCID site and allow you to create a new iD or authenticate a pre-existing iD in Editorial Manager. Please see the following video for instructions on linking an ORCID iD to your Editorial Manager account: https://www.youtube.com/watch?v=_xcclfuvtxQ"". 

Reviewers' comments:

Reviewer's Responses to Questions

**Comments to the Author**

1. Is the manuscript technically sound, and do the data support the conclusions?

Reviewer #1: Yes

2. Has the statistical analysis been performed appropriately and rigorously? 

Reviewer #1: Yes

3. Have the authors made all data underlying the findings in their manuscript fully available?

Reviewer #1: Yes

4. Is the manuscript presented in an intelligible fashion and written in standard English?

Reviewer #1: Yes

5. Review Comments to the Author

Reviewer #1: This is interesting topic, that has already been covered in the literature substantially. The authors are bringing back the idea that time to peak torque could be equally or even more important than maximal torque output.

I have several important remarks, that I would like authors to address. They are listed below.

Why did you decide to test from 90-0°. This may influence test results due to fatigue, especially as you had quite an extensive testing protocol in regard to reps.

Name the model of the dynamometer.

Do you think 30sec rest between speeds is enough?

How come you decided to use average and not peak torque values? Even more, you have obviously calculated average from both legs which is odd and provide any reference for that. As isokinetic strength testing is my area of expertise I must say that you should use peak torque (not average) and dominant side (in basketball one used to push off in two-step).

Also I think that you are lacking jump without arm swing, as many studies have confirmed that arm swing may significantly influence the height of the jump and this in turns may explain why you didn’t find significant correlations with knee extensor strength. OK I see that you were not using force platform but Vertec - I hope this is in limitations.

You body weight normalized values are actually in %. Add % or change the decimal point.

You are mixing anglosaxon and metric system. Check Plos instructions about units of measure.

You are repeating some information from the Table in the text. Don’t do so.

You linear regression model is prone to bias. Have you been doing multiple regression approach as well? Adding peak torque instead of average may significantly influence correlations as well as whole model.

Can you explain in more details this assumption “It was hypothesized that time to peak torque at the highest °/ speed of 300 s would have the strongest correlation with VJ performance.” It is maximal strength that matters for VJ, and that one is best measured at low concentric velocities. If you wanted time include test power instead of time to peak torque

In the discussion section you are talking about the peak torque, but instead you were actually using average - see above.

For a study showing really low R2 albeit significant I would tune down the suggestion in regard to that finding.

6. PLOS authors have the option to publish the peer review history of their article (what does this mean? ). If published, this will include your full peer review and any attached files.

**Do you want your identity to be public for this peer review?** For information about this choice, including consent withdrawal, please see our Privacy Policy .

Reviewer #1: No

---

## [Author Response · Author response to Decision Letter 1]

19 Jan 2024

Dear Reviewers,

Below are your concerns addressed:

1. Why did you decide to test from 90-0 degrees?

Thank you for the comment. Range of motion limitations were set based on prior protocol and studies. We are not aware of any literature that would suggest that this many influence test results in regard to reps but would be happy to include any literature that does suggest that it may be a limitation.

2. Name the Model of the Dynamometer

This is addressed in the methods section. It is a Biodex Dynamometer.

3. Do you think 30 seconds rest between speeds is enough?

Thank you for the comment. We consider a 30 second break between speeds to be sufficient. Prior studies have concluded that there was no significant difference in knee extension or knee flexion peak torque when comparing work to rest ratios. We are not aware of any literature that would suggest that it isn’t but would be happy to include any literature that does suggest it isn’t enough and mention it in a limitations section.

4. How come you decided to use an average and not peak torque values?

Thank you for the comment. We consider the average value to be more stable of a variable than peak torque as the average value is based on all of the repetitions rather than just one repetition for the peak torque variable.

5. Even more, you have obviously calculated average from both legs which is odd and provide

any reference for that. As isokinetic strength testing is my area of expertise I must say that you should use peak torque (not average) and dominant side (in basketball one used to push off in two-step).

Thank you for your expert comment, we appreciate your expertise. Please see above our rational for using the average peak torque. The task we utilized in this study was a two-legged countermovement jump rather than a running vertical jump. Given this was a two-legged task, we consider the variables included in the statistical analysis to be appropriate for the manuscript.

6. Also, I think that you are lacking jump without arm swing, as many studies have confirmed that arm swing may significantly influence the height of the jump and this in turns may explain why you didn’t find significant correlations with knee extensor strength. OK I see that you were not using force platform but Vertec - I hope this is in limitations

Thank you for the comment. We only used the Vertec for the jump height which requires the use of the hands/arms. We will mention this in the limitations section with the follow language – “The current study allowed the subjects to use their arms/hands during the jumps which is necessary based on the used of the Vertec. This could have affected the ability to find correlations between jump height and knee extensor strength.”

7. Your body weight normalized values are actually in %. Add % or change the decimal point. You are mixing anglosaxon and metric system. Check Plos instructions about units of measure. You are repeating some information from the Table in the text. Don’t do so.

Thank you for your comment. The data presented in Tables 2 and 3 are calculated as a ratio of peak torque normalized to body weight and are presented as such. All units are updated to be correctly in the metric system (torque in Nm, weight as kg, and time in ms). Vertical jump was originally reported in inches and has been corrected to centimeters.

8. Your linear regression model is prone to bias. Have you been doing multiple regression approach as well? Adding peak torque instead of average may significantly influence correlations as well as whole model.

Thank you for the comment. Please see previous comments/response to this variable. We have included a statement in our limitations section.

9. Can you explain in more details this assumption “It was hypothesized that time to peak torque at the highest °/ speed of 300 s would have the strongest correlation with VJ performance.” It is maximal strength that matters for VJ, and that one is best measured at low concentric velocities. If you wanted time include test power instead of time to peak torque

Thank you for the comment. We created this hypothesis because the speed of 300 degrees/second likely is closer to the angular velocity of the actual movement.

10. In the discussion section you are talking about the peak torque, but instead you were actually using average - see above.

Thank you for the comment. We have updated the language in the discussion to reflect “average peak torque” as the variable of discussion.

11. For a study showing really low R2 albeit significant I would tune down the suggestion in regard to that finding.

Thank you for the comment.

Thank you so much for your review.

---

## [Decision Letter · Decision Letter 1]

20 Feb 2024

PONE-D-23-32453R1Examining Isokinetic Knee Peak Torque and Time to Peak Torque as Predictors of Vertical Jump Height in Division I Men’s Basketball PlayersPLOS ONE

Dear Dr. Westwood,

Thank you for submitting your manuscript to PLOS ONE. After careful consideration, we feel that it has merit but does not fully meet PLOS ONE’s publication criteria as it currently stands. Therefore, we invite you to submit a revised version of the manuscript that addresses the points raised during the review process.

**ACADEMIC EDITOR: **

Dear Authors, two experts in the field reviewed your manuscript reporting some minor issues you should consider during the revision process. Please submit your revised manuscript by Apr 05 2024 11:59PM. If you will need more time than this to complete your revisions, please reply to this message or contact the journal office at plosone@plos.org . Please include the following items when submitting your revised manuscript:

We look forward to receiving your revised manuscript.

Kind regards,

Emiliano Cè

Academic Editor

PLOS ONE

Journal Requirements:

Reviewers' comments:

Reviewer's Responses to Questions

**Comments to the Author**

1. If the authors have adequately addressed your comments raised in a previous round of review and you feel that this manuscript is now acceptable for publication, you may indicate that here to bypass the “Comments to the Author” section, enter your conflict of interest statement in the “Confidential to Editor” section, and submit your "Accept" recommendation.

Reviewer #2: All comments have been addressed

Reviewer #3: All comments have been addressed

2. Is the manuscript technically sound, and do the data support the conclusions?

Reviewer #2: Yes

Reviewer #3: Yes

3. Has the statistical analysis been performed appropriately and rigorously? 

Reviewer #2: Yes

Reviewer #3: Yes

4. Have the authors made all data underlying the findings in their manuscript fully available?

Reviewer #2: Yes

Reviewer #3: Yes

5. Is the manuscript presented in an intelligible fashion and written in standard English?

Reviewer #2: Yes

Reviewer #3: No

6. Review Comments to the Author

Reviewer #2: Congrulations! Your manuscript is ready to publish. Thank ou so much for your all response.

Best Regards

Reviewer #3: PONE-D-23-32453R1

Examining Isokinetic Knee Peak Torque and Time to Peak Torque as Predictors of Vertical Jump Height using Vertec in Division I Men’s Basketball Players

Dear Authors

Thank you for the opportunity to review the article PONE-D-23-32453R1 - Examining Isokinetic Knee Peak Torque and Time to Peak Torque as Predictors of Vertical Jump Height in Division I Men’s Basketball Players for the journal PLOS ONE.

The work is generally relevant to sports science, particularly basketball, falling within the journal's scope.

Therefore, the article should be accepted for publication after being adjusted according to the comments below.

Best Regards

General comments

The use of Vertec is a limitation.

The protocol used could (should) have ALSO considered 2 VJs that are part of Bosco's protocol: squat jump and countermovement jump. The work was left with almost no limitations, and the findings would undoubtedly have a more significant scientific impact.

However, taking into account the rationale behind the modelling, it can be seen that the objective is focused, specifically, on using equipment without technological limitations that may be available at training locations. Thus, performance in VJ with vortex can, at any time, be interpreted in light of this work. However, it is essential to emphasise that the results of this study are only valid for these conditions.

However, the use of the methodology stated is also an advantage due to the standardisation of the VJ movement, but because it resembles the VJ most performed by athletes in the studied modality.

The question is: Which studies use the same methodology (BIODEX vs VERTEC)? The work would be significantly improved if the reference bibliography (mainly) focused on protocols with the same studied vertical jump (Vertec).

Specific comments

Title

Suggestion: Examining Isokinetic Knee Peak Torque and Time to Peak Torque as Predictors of Vertical Jump Height using Vertec apparatus in Division I Men’s Basketball Players

Discussion

L178-180: “Overall, the results of this study did not support this hypothesis, as none of the predictor variables showed a significant correlation with VJ height.” // No. The results support the hypothesis, i.e., time to peak torque at the highest speed of 300 °/s would have the strongest correlation with VJ performance. Review this sentence to clarify.

L197-198: “This experiment failed to find a significant correlation between peak torque and VJ height, and this may be explained by the role of the hip joint during the VJ.” // You could also reference the VJ assessment methodology (i.e., using VERTEC, which is not a standard vertical jump test like the SJ, CMJ or ABK).

L2002: “semimembranosus and rectus femoris” // The muscles semitendinosus and (long head of) biceps femoris (booth from the posterior region) are missing (they are also biarticular muscles – hip and knee).

Pratical applications

This section is a mix of discussion and practical applications. I believe that to improve this section considerably, it can (should) be reformulated (i.e., present an objective and direct text in which the reasoning - and respective references - would only be considered in the “discussion” section).

7. PLOS authors have the option to publish the peer review history of their article (what does this mean? ). If published, this will include your full peer review and any attached files.

**Do you want your identity to be public for this peer review?** For information about this choice, including consent withdrawal, please see our Privacy Policy .

Reviewer #2: No

Reviewer #3: No

---

## [Author Response · Author response to Decision Letter 2]

9 Apr 2024

Dear Reviewers,

Below are your concerns addressed:

1. The use of Vertec is a limitation.

Thank you for the comment. The Vertec has been shown to be a valid and reliable measure for lower extremity testing. Use of the Vertec was based on prior protocol and studies. We are not aware of any significant limitations of Vertec that may influence test results but would be happy to include any literature that does suggest that it may be a limitation.

2. The protocol used could (should) have ALSO considered 2 VJs that are part of Bosco's protocol: squat jump and countermovement jump. The work was left with almost no limitations, and the findings would undoubtedly have a more significant scientific impact. However, taking into account the rationale behind the modelling, it can be seen that the objective is focused, specifically, on using equipment without technological limitations that may be available at training locations. Thus, performance in VJ with vortex can, at any time, be interpreted in light of this work. However, it is essential to emphasise that the results of this study are only valid for these conditions. However, the use of the methodology stated is also an advantage due to the standardisation of the VJ movement, but because it resembles the VJ most performed by athletes in the studied modality. The question is: Which studies use the same methodology (BIODEX vs VERTEC)? The work would be significantly improved if the reference bibliography (mainly) focused on protocols with the same studied vertical jump (Vertec).

Thank you for the comment, we appreciate your expertise. We considered the choice of a two-legged countermovement jump and variables included in the statistical analysis appropriate for the manuscript and this study population. We considered all reference bibliography appropriate for the discussion and relevant for comparison.

Specific comments

1. Title: Suggestion: Examining Isokinetic Knee Peak Torque and Time to Peak Torque as Predictors of Vertical Jump Height using Vertec apparatus in Division I Men’s Basketball Players

Thank you for your suggestion. We believe the current title is appropriate for the contents of the paper.

2. Discussion

L178-180: “Overall, the results of this study did not support this hypothesis, as none of the predictor variables showed a significant correlation with VJ height.” // No. The results support the hypothesis, i.e., time to peak torque at the highest speed of 300 °/s would have the strongest correlation with VJ performance. Review this sentence to clarify.

Thank you for the comment. Our hypothesis was specific to the predictor variables; therefore, the results did not support this original hypothesis. We believe the following lines clarify that the regression analysis showed significance.

L197-198: “This experiment failed to find a significant correlation between peak torque and VJ height, and this may be explained by the role of the hip joint during the VJ.” // You could also reference the VJ assessment methodology (i.e., using VERTEC, which is not a standard vertical jump test like the SJ, CMJ or ABK).

Thank you for the suggestion. We addressed limitations in methodology in terms of the use of arm swing and generalizability of the subjects. As addressed above, we utilized VERTEC based on prior protocol and studies which showed reliability of its use.

L2002: “semimembranosus and rectus femoris” // The muscles semitendinosus and (long head of) biceps femoris (booth from the posterior region) are missing (they are also biarticular muscles – hip and knee).

Thank you for the comment, we will add these additional muscles into the sentence to properly reflect the biarticular muscles.

3. Practical applications

This section is a mix of discussion and practical applications. I believe that to improve this section considerably, it can (should) be reformulated (i.e., present an objective and direct text in which the reasoning - and respective references - would only be considered in the “discussion” section).

Thank you for your comments. We wrote this section such that the object is the “focus on rapid generation of torque in addition to traditional strengthening,” followed up with respective references in plyometrics, Olympic weightlifting, and male soccer players.

Thank you for your review.

---

## [Decision Letter · Decision Letter 2]

30 Apr 2024

Examining Isokinetic Knee Peak Torque and Time to Peak Torque as Predictors of Vertical Jump Height in Division I Men’s Basketball Players

PONE-D-23-32453R2

Dear Dr. Westwood,

We’re pleased to inform you that your manuscript has been judged scientifically suitable for publication and will be formally accepted for publication once it meets all outstanding technical requirements.

Kind regards,

Emiliano Cè

Academic Editor

PLOS ONE

Additional Editor Comments (optional):

Reviewers' comments:

Reviewer's Responses to Questions

**Comments to the Author**

1. If the authors have adequately addressed your comments raised in a previous round of review and you feel that this manuscript is now acceptable for publication, you may indicate that here to bypass the “Comments to the Author” section, enter your conflict of interest statement in the “Confidential to Editor” section, and submit your "Accept" recommendation.

Reviewer #3: All comments have been addressed

2. Is the manuscript technically sound, and do the data support the conclusions?

Reviewer #3: Yes

3. Has the statistical analysis been performed appropriately and rigorously? 

Reviewer #3: Yes

4. Have the authors made all data underlying the findings in their manuscript fully available?

Reviewer #3: No

5. Is the manuscript presented in an intelligible fashion and written in standard English?

Reviewer #3: Yes

6. Review Comments to the Author

Reviewer #3: Dear Authors

Thank you for your responses to the comments presented in the initial review and changes presented in the article PONE-D-23-32453R2 - Examining Isokinetic Knee Peak Torque and Time to Peak Torque as Predictors of Vertical Jump Height in Division I Men’s Basketball Players.

Congratulations.

7. PLOS authors have the option to publish the peer review history of their article (what does this mean? ). If published, this will include your full peer review and any attached files.

**Do you want your identity to be public for this peer review?** For information about this choice, including consent withdrawal, please see our Privacy Policy .

Reviewer #3: **Yes: ** Luís Miguel Massuça

---

## [Editor Report · Acceptance letter]

PONE-D-23-32453R2

PLOS ONE

Dear Dr. Westwood,

I'm pleased to inform you that your manuscript has been deemed suitable for publication in PLOS ONE. Congratulations! Your manuscript is now being handed over to our production team.

Kind regards,

on behalf of

Prof. Emiliano Cè

Academic Editor

PLOS ONE